# Densazalin, a New Cytotoxic Diazatricyclic Alkaloid from the Marine Sponge *Haliclona densaspicula*

**DOI:** 10.3390/molecules26113164

**Published:** 2021-05-25

**Authors:** Buyng Su Hwang, Yong Tae Jeong, Sangbum Lee, Eun Ju Jeong, Jung-Rae Rho

**Affiliations:** 1Department of Oceanography, Kunsan National University, Jeonbuk 54150, Korea; hwang1531@nnibr.re.kr (B.S.H.); sblee08@kunsan.ac.kr (S.L.); 2Nakdonggang National Institute of Biological Resources, Gyeongbuk 37242, Korea; ytjeong@nnibr.re.kr; 3Department of Plant & Biomaterials Science, Gyeongsang National University, Jinju 52725, Korea

**Keywords:** *Haliclona densaspicula*, densazalin, 1,3-alkylpyridine, cytotoxic alkaloid

## Abstract

Densazalin, a polycyclic alkaloid, was isolated from the marine sponge *Haliclona densaspicula* collected in Korea. The complete structure of the compound was determined by spectroscopic methods, including 1D and 2D nuclear magnetic resonance techniques, high-resolution mass spectrometry, and comparison of the calculated and measured electronic circular dichroism spectra. Densazalin possesses a unique 5,11-diazatricyclo[7.3.1.0^2,7^]tridecan-2,4,6-triene moiety, which is connected by two linear carbon chains. This compound was derived from the biogenetic precursor bis-1,3-dialkylpyridnium. Densazalin exhibited cytotoxic activity on two human tumor cell lines (AGS and HepG2) in the Cell Counting Kit-8 (CCK-8) bioassay, with IC_50_ values ranging from 15.5 to 18.4 μM.

## 1. Introduction

Marine sponges are prolific sources of natural products that show significant bioactivity and possess noble chemical structures that have contributed to the discovery and development of pharmaceutical drugs [1,2]. In sponges from the genus *Haliclona*, bioactive secondary metabolites including sterols [3], cyclic peptides [4], macrolides [5], polyacetylenes [6], and alkaloids such as 1,3-alkylpyridines [7] have been identified.

In particular, structurally diverse polycyclic alkaloids with two heterocyclic nitrogens have been clarified from Haplosclerid sponges [8]. Haplosclerid sponges, notably those from the genera *Haliclona*, *Xestospongia*, and *Amphimedon* spp., are rich sources of structurally complex and cytotoxic alkaloids derived from 3-alkylpyridines or their reduction products [8].

A variety of alkaloids, such as halitoxin [9], manzamines [10], manzamine B and C [11], haliclamines A and B [12], 1,2,3,4-tetrahydro-8-hydroxymanzamines [13], and halicyclamine A, were identified from the genus *Haliclona*. The structure elucidation of these alkaloids with complex skeletons has been extremely challenging. In 1992, a retrobiosynthetic scheme for some of their complex frameworks was clarified by Baldwin and Whitehead [14]. Halicyclamine A represents a tetracyclic alkaloid skeleton biogenetically related to the xestocyclamine/ingenamine class of alkaloids [15].

The cyclic bis-1,3-dialkylpyridinium alkaloids, such as the sarains [16], manzamines [9], densanins [17], and haliclonacyclamines [11], have been recognized as biogenetic precursors to unique polycyclic nitrogenous metabolites with diverse biological activities. A variety of biological activities of *Haliclona* spp., including antimicrobial [18,19], antibacterial [20], antifungal [21,22], hemagglutination [23,24], anti-cancer [25,26,27,28], and anti-inflammatory activities [17,29], have been reported.

In a previous study, our research team reported new macrocyclic pyrrole alkaloids, densanins A and B, with potent anti-inflammatory activity, from *Haliclona densaspicula* [17]. In our continuing search for novel bioactive metabolites from marine species, the methanolic extract of *H. densaspicula* showed considerable cytotoxicity against human tumor cell lines (AGS and HepG2) when subjected to the Cell Counting Kit-8(CCK-8) bioassay. Bioassay-guided separation and chemical investigation of the extract using successive column chromatographies over RP-18 silica gel and Sephadex LH-20, followed by semi-preparative LC, led to the isolation of a new polycyclic alkaloid, densazalin (**1**), which could be derived from a cyclic bis-1,3-dialkylpyridinium. The extract in the previous study provided two alkaloids, densanins A and B, showing inhibition activity against LPS-induced NO production in BV-2 microglial cells [17]. Here, we report the structure determination of the new skeleton of **1** by the spectroscopic methods, quantum calculations and strong cytotoxic activity.

## 2. Results and Discussion

Densazalin (**1**) was found to have the molecular formula of C_32_H_47_N_2_^+^ [M+] from its positive HR-ESITOF mass spectrum and its ^13^C NMR spectrum, indicating 11 degrees of unsaturation. The UV and IR spectra displayed absorption peaks at 230, 270 nm and 2360, 1595 cm^−1^, respectively, of which 270 nm and 1595 cm^−1^ signals indicated a pyridine functional group. The ^1^H NMR spectrum was characterized by the absence of methyl signals, as well as severely overlapped signals in the upfield region which corresponded to an aliphatic chain, while downfield-shifted protons indicated an aromatic group. Based on the ^13^C NMR and HSQC spectra, **1** consisted of 18 methylenes containing 4 nitrogen-bearing carbons (δ_C_ 59.2, 60.3, 62.8, 66.0), 11 methines, and 3 non-protonated carbons (δ_C_ 38.2, 144.5, 163.6) (Table 1). **1** possessed a 1,3,4-trisubstituted pyridinium ring, which was deduced by the aromatic protons at δ_H_ 7.92 (H-3), 8.65 (H-4), and 8.59 (H-6), and the two downfield-shifted carbons at δ_C_ 141.5 (C-4) and 145.1 (C-6) with large one-bond heteronuclear coupling constants, ^1^*J*_CH_ = 194 and 189 Hz, respectively. These were corroborated by the HMBC cross-peaks of the two protons (H-4 and H-6) with the nitrogen-bearing methylene carbon at δ_C_ 62.8 (C-1′). Together with the pyridinium ring, three additional double bonds accounted for seven out of the 11 degrees of unsaturation, which suggested the presence of four rings in **1**.



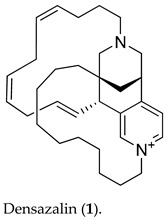



Detailed interpretation of 1D and 2D NMR (COSY, TOCSY, HSQC, and HMBC, in Appendix A) spectra established that **1** was composed of a 5,11-diazatricyclo[7.3.1.0^2,7^]tridecan-2,4,6-triene moiety [30] and two independent linear carbon chains A and B, as shown in Figure 1. Based on the three-carbon connectivity from the COSY correlations of H-12/H-1 and H-1/H-13, the structure of the 5,11-diazatricyclo[7.3.1.0^2,7^]tridecan-2,4,6-triene was identified by the HMBC correlations between H-12b and the nitrogen-bearing carbon (C-10), and the two protons (H-10a and H-13b) and the quaternary carbon (C-9), with additional HMBC correlations of H-8/C-9, C-8/C-7, H-12a/C-2, and H-13a/C-2. HMBC cross-peaks of H-8/C-10, H-8/C-13, H-8/C-6, H-8/C-2, H-3/C-7, H-6/C-4, H-6/C-2, and H-4/C-2 supported this partial structure. One of the two independent carbon chains was assigned as Δ^1,4,7^-undecatriene by the sequential COSY and TOCSY correlations between the relatively well-resolved proton signals in the chain. The remaining 10 methylene carbons constituted the other chain, which was assigned as the *n*-decan functional group. The three partial structures could be assembled on the basis of the HMBC correlations. C-1′ and C-10′ at the end of chain A were linked to N-5 and C-9 of the diazatricyclic ring, respectively, as evidenced by the key HMBC correlations of H-4/C-1′, H-6/C-1′, and H-8/C-10′. Very similarly, C-1″ and C-11″ at the terminus of chain B were connected to N-11 and C-8 on the diazatricyclic ring, respectively, from the HMBC correlations of H-1″/C-10, H-1″/C-12, and H-11″/C-8. Accordingly, the positively charged gross structure of **1** was completed, being consistent with the molecular formula.

The structure of densazalin, containing three chiral centers and three double bonds, was configured by analysis of the NOESY spectrum and the proton coupling constants. NOE correlations of H-1/H-12b and H-1/H-13 established that H-1 is in an equatorial position of the tetradehydropyran ring, and the perpendicular placement of the pyridinium ring to the tetrahydropyran ring was supported by the NOE correlations between H-3 and H-12b. Based on this configuration, the NOE cross-peaks of H-6/H-8 and H-8/H-13b led to the configuration of C-8 and C-9 as *R** and *R** forms, respectively (Figure 2). This configuration allowed us to observe the NOE between H-11″ and H-3″b at a long distance. Moreover, the abnormal upfield shift of H-4’a was caused by the location of this proton above the pyridinium ring. The geometries of the double bonds in chain A were assigned as 4*Z*, 6*Z*, and 10*E* by the NOE correlations of H-2″/H-5″, H-6″/H-9″ and the large coupling constant (*J* = 15.9 Hz) between H-10″ and H-11″. The absolute stereochemistry of **1** was elucidated by the comparison of the quantum chemically calculated electronic circular dichroism (ECD) with the measured spectrum, as shown in Figure 3. After generating the input structure of **1** as determined by the NOE correlation studies, a conformational search was performed using molecular mechanics with MMFF, and eight conformers with low energies were determined within a 12 kJ/mol threshold. Geometry optimization of each conformer was performed at the B3LYP/6-31G(d) level, and then their single-point energies were calculated at the B3LYP/6-311G+(2d,p) level to obtain their respective Boltzmann populations at 298 K (Appendix A). The calculations of the ECD spectra for all populated conformers were performed at the B3LYP/6-31G(d,p) level [31].

A plausible biosynthesis of **1** would be similar to that suggested for the formation of the halicyclamine structure from a 1,3-dialkaylpyridinium [15] (Figure 4): the two pyridiniums rings formed a xestocyclamine skeleton by Diels–Alder cyclization [32], and subsequent cleavage of the C-7/C-10 bond, formation of the C-8/C-9 bond, and reduction lead to a diazatricyclic ring.

Densazalin showed cytotoxicity against the AGS and HepG2 cell lines by CCK-8 assay at above a concentration of 10 μM (Figure 5). IC_50_ values for **1** were measured as 15.5 and 18.4 μM for the AGS and HepG2 cell lines, respectively. This value indicates moderate cytotoxicity compared to that of several marine alkaloids. For example, deoxytopsentin isolated from the marine sponge *Spongsorites* sp. shows 4.0 μM for AGS and 10.2 μM for HepG2 [33].

## 3. Materials and Methods

### 3.1. Instrumentation

Optical rotation was measured in a cell with 5 cm path length on a JASCO P-1010 polarimeter (Jasco, Easton, MD, USA) with MeOH as blank. IR and UV spectra were recorded on a JASCO FT-IR 4100 spectrometer (Jasco, Easton, MD, USA) and a Varian Cary 50 UV–visible spectrophotometer (Agilent, Santa Clara, CA, USA), respectively. High-resolution (HR) electrospray ionization (ESI) mass data were obtained using a SCIEX X500R mass spectrometer (SCIEX, Framingham, MA, USA). All nuclear magnetic resonance (NMR) spectra were recorded on a Varian VNMRS 500 NMR spectrometer (Varian, Palo Alto, CA, USA) operating at 500 MHz (^1^H) and 125 MHz (^13^C). All 1D and 2D NMR spectra were measured in methanol-d_4_ solvent at 25 °C and referenced at 3.3 ppm (^1^H) and 49.0 ppm (^13^C) for the solvent peak. The parameters used for 2D NMR spectra were as follows; The gradient COSY spectrum was collected with a spectral width of 4600 Hz in a 1024(*t*1) × 2048 (*t*2) matrix applying a pulse gradient of 1 ms duration with a strength of 10 G/m and processed with a sine-bell function. The gradient HSQC data were obtained in a 128 (*t*1) × 1024 (*t*2) matrix with ^1^*J*_CH_ = 140 Hz and processed in a 256 (*t*1) × 1024 (*t*2) matrix by a linear prediction method for a higher resolution. The gradient HMBC experiment was optimized for a long-range coupling constant of 8 Hz. The NOESY experiment was carried out with a mixing time of 250 ms. Semi-preparative liquid chromatography (Prep-LC) was performed on an Agilent 1200 pump (Agilent, Santa Clara, CA, USA) equipped with a DAD detector. Isolation of compounds was performed with an RP C_18_ silica gel 60 (Merck, Darmstadt, Germany) or Sephadex LH-20 (Pharmacia, Uppsala, Sweden).

### 3.2. Material

A specimen of *Haliclona densaspicula* (voucher number 08K-11) was collected by scuba divers off Keomun Island, South Korea, in 2008. The sponge was massive and easily broken like bread. The surface was smooth and had several oscules. When living, the color was brown. In this study, the aliquot extracted in 2012 was used.

### 3.3. Extraction and Isolation

The methanolic extract *H. densaspicula* was partitioned between dichloromethane (DCM) and distilled water. The organic fraction was repartitioned into *n*-hexane and 15% aqueous MeOH. The aqueous MeOH fraction (ca 4.5 g) was subjected to reversed-phase silica gel flash column chromatography, eluting with solvents of decreasing polarity (MeOH:H_2_O = 5:5 → 6:4 → 7:3 → 8:2 → 9:1 → 100% MeOH → 100% acetone) to give seven fractions (MR1~MR7). A Cell Counting Kit-8 (CCK-8) bioassay was performed for each fraction to select a fraction with potent cytotoxicity using human tumor cell lines, AGS and HepG2.

The active fraction, the MR5 fraction (430 mg), was chromatographed on an LH-20 column and eluted with 100% MeOH to yield five fractions (M1~M5). Subfraction M3 (115 mg) was separated by semi-preparative reversed-phase HPLC with a UV detector, using a YMC ODS-A column (250 mm × 10 mm i.d., 5 μm) with a solvent system of MeCN and H_2_O [2 (MeCN): 8 (H_2_O) → 10: 0] in a flow rate of 2 mL/min to yield two mixed fractions. Densazalin (**1**) (2.3 mg) was purified by reversed-phase HPLC using a phenomenex C6-phenyl column (250 mm × 10 mm i.d., 5 μm), eluting with a solvent system of 60% MeOH and 40% H_2_O. The compound was produced at a retention time of 50 min.

Densazalin (**1**). Yellow oil. [α]-18.2 (c 0.04, MeOH). UV (MeOH) λ_max_ (log ε): 203 (4.5), 230 (4.1), 270 (3.7) nm. IR (film) ν_max_: 2938, 2360, 1595 cm^−1^. ^1^H (500 MHz) and ^13^C (125 MHz) NMR data, see Table 1. HR-ESITOF MS (positive-ion mode) *m*/*z*: 459.3716 [M]^+^ (calculated for C_32_H_47_N_2_^+^, 459.3734).

### 3.4. Cell Culture and Cytotoxicity Assay

The in vitro cytotoxicity was measured using a Cell Counting Kit-8 (CCK-8) (DOJINDO, Kumamoto, Japan) on AGS cells (human gastric adenocarcinoma cell line), with HepG2 cells (human hepatocellular carcinoma cell line) as a vehicle group. Cell lines were obtained from the American Type Culture Collection (ATCC, Rockville, MD, USA) and were cultured at 37 °C in a 5% CO_2_ incubator (Thermo, Waltham, MA, USA). DMEM supplemented with 10% fetal bovine serum and 1% penicillin/streptomycin was used.

Each cell line was seeded onto 96-well plates at 1 × 10^5^ cells per well and cultured overnight for use in the experiments. The cells were cultured for 24 h in medium containing various concentrations of denazalin (**1**) for use in the experiments. Next, 10 μL CCK-8 was added to the cells and incubated for 2 h, and optical density was measured at 450 nm using a Cytation 3 microplate reader (Biotek, Winooski, VT, USA). The cytotoxicity was calculated as % relative to the vehicle group. Dose–response curves were established for the sample and the minimum concentration sufficient to reduce the cell viability by 50% (IC_50_) was calculated.

### 3.5. Statistical Analysis

SPSS 25 software was used for statistical analysis. All experiments described were performed at least three times or more. Data were expressed as the mean ± SD. Significant differences between the means of two groups were determined by a Student’s *t*-test. A *p* value less than 0.05 was considered as statistically different.

### 3.6. Calculation of ECD Spectrum of ***1***

The configurational structure of densazalin determined by NMR spectroscopy was used as an input for a conformational search of **1**. The conformational search was performed by Spartan 18 software (Wavefunction Inc., Irvine, CA, USA), which calculates using molecular mechanics. Eight conformers with low energies were selected within a 12 kJ/mol threshold. Each conformer was optimized by the DFT method at the B3LYP/6-31G(d) level using the Gaussian 16 program (Gaussian. Inc., Wallingford, CT, USA). Following this procedure, the ECD spectrum of each conformer was calculated by the TD-DFT method at the B3LYP/6-31G(d,p) level with the PCM model in methanol solvent. The weights of the conformers by Boltzmann distribution were obtained from the calculation of the single-point energies of eight conformers at the B3LYP/6-311G+(2d,p) level.

## 4. Conclusions

The structure of a new compound isolated from the marine sponge *H. densaspicular*, densazalin (**1**), was completely determined by spectroscopic methods and quantum mechanical calculations. **1** was deduced to be derived from a cyclic bis-1,3-dialkylpyridinium species, similar to densanins A and B isolated from the same extract. According to our best knowledge, the 5,11-diazatricyclo[7.3.1.0^2,7^]tridecan-2,4,6-triene moiety in **1** has been synthesized [30], but this is the first report of its occurrence in a natural product. With regard to bioactivity, **1** showed toxicity against two tumor cells lines; these anticancer effects are similar to others reported for cytotoxicity evaluations of the alkaloid compounds from the genus *Haliclona* [34,35]. Finally, densazalin, with its high toxicity against the AGS and HepG2 cell lines, may be especially promising for developing an effective drug against melanoma and ovarian cancer in this regard.

## Figures and Tables

**Figure 1 molecules-26-03164-f001:**
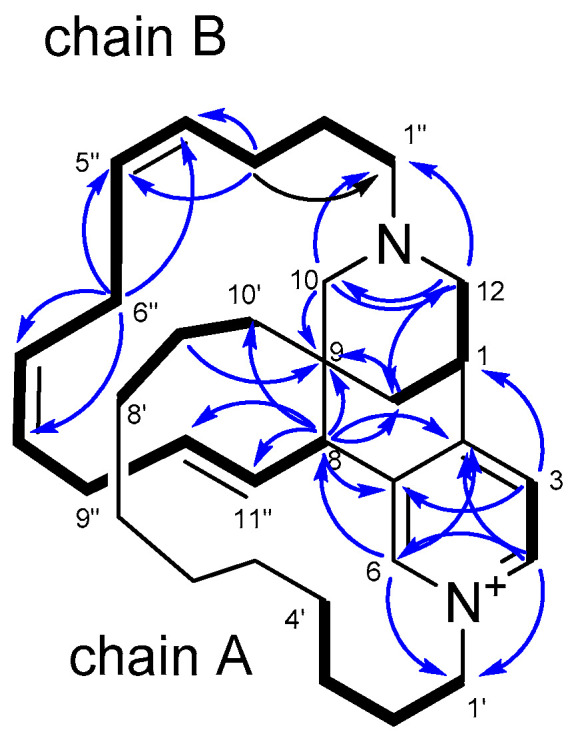
COSY and key HMBC correlations in **1**.

**Figure 2 molecules-26-03164-f002:**
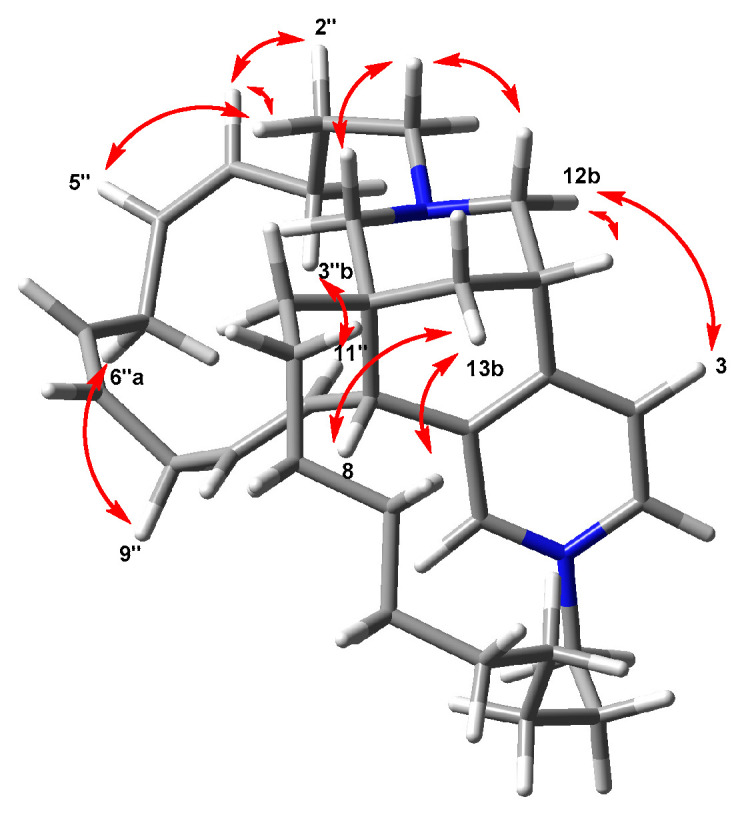
Selected NOE correlations.

**Figure 3 molecules-26-03164-f003:**
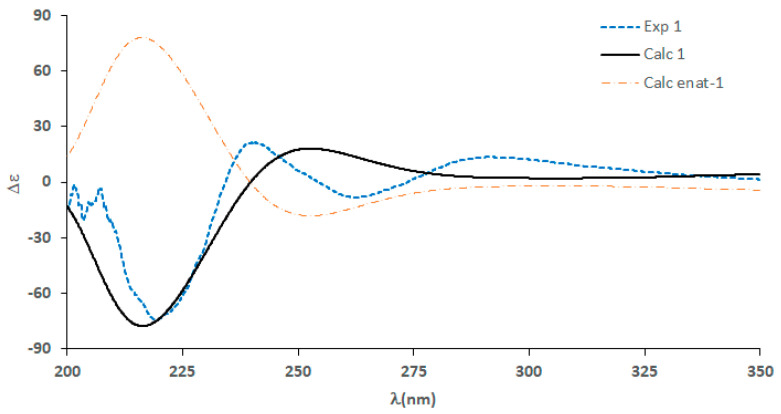
Experimental and calculated ECD spectra of **1**.

**Figure 4 molecules-26-03164-f004:**
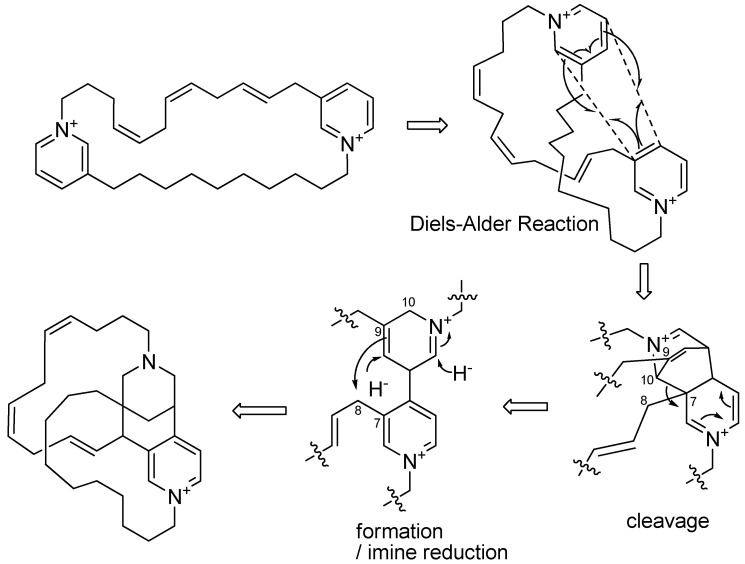
Plausible biosynthesis of **1**.

**Figure 5 molecules-26-03164-f005:**
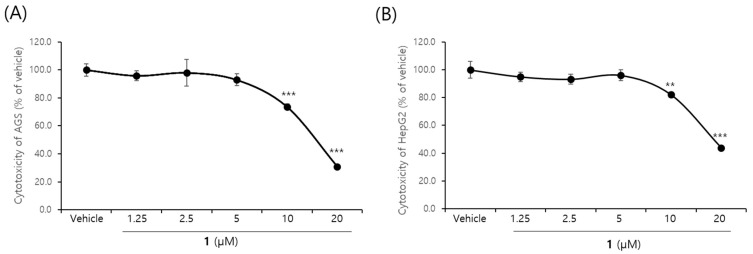
Dose–response curves of **1** on AGS (**A**) and HepG2 (**B**) carcinoma cell lines. Cells were cultured with 0–20 μM of **1** for 24 h. Each value is expressed as the means ± SD of triplicate determinations. ** *p* < 0.01, *** *p* < 0.001 versus vehicle group.

**Table 1 molecules-26-03164-t001:** ^1^H (500 MHz) and ^13^C NMR (125 MHz) data for compound **1** in CD_3_OD (δ in ppm, *J* values in parentheses).

no.	δ_C_	δ_H,_ mult (*J* Hz)
1	38.4, CH	3.35, m
2	163.6, C	
3	128.6, CH	7.92, d(6.1)
4	141.5, CH	8.65, d(6.1)
6	145.1, CH	8.59, s
7	144.5, C	
8	46.4, CH	3.77, d(8.8)
9	38.2, C	
10	66.0, CH_2_	^a^ 1.59, d(11.3); ^b^ 2.87, brd(11.3)
12	60.3, CH_2_	^a^ 2.39, dd(11.6, 3.2); ^b^ 3.06, d(11.6)
13	35.6, CH_2_	^a^ 1.58, dd(13.2, 2.7); ^b^ 2.02, brd(13.2)
1′	62.8, CH_2_	^a^ 4.54, td(12.5, 3.2); ^b^ 4.72, dt(12.5, 3.9)
2′	30.9, CH_2_	^a^ 1.72, m; ^b^ 2.15, m
3′	26.2, CH_2_	^a^ 1.24, m; ^b^ 1.44, m
4′	29.5, CH_2_	^a^ 0.07, m; ^b^ 1.07, m
5′	29.1, CH_2_	^a^ 1.17, m; ^b^ 1.25, m
6′	30.3, CH_2_	1.12, m
7′	31.0, CH_2_	1.19, m
8′	30.5, CH_2_	1.24, m
9′	23.1, CH_2_	^a^ 1.36, m; ^b^ 1.67, m
10′	35.9, CH_2_	^a^ 1.23, m; ^b^ 1.57, m
1″	59.2, CH_2_	^a^ 2.19, m; ^b^ 2.28, dd(7.6, 3.7)
2″	27.9, CH_2_	1.41, dd(7.6, 3.4)
3″	27.4, CH_2_	^a^ 1.67, m; ^b^ 2.17, m
4″	132.4, CH	5.29, td(10.9, 5.1)
5″	128.9, CH	5.48, m
6″	26.3, CH_2_	^a^ 2.65, m; ^b^ 2.96, m
7″	130.9, CH	5.41, dt(11.3, 6.1)
8″	128.0, CH	5.54, m
9″	30.6, CH_2_	2.92, m
10″	134.9, CH	6.10, dt(15.9, 6.1)
11″	131.8, CH	5.94, dd(15.9, 9.1)

^a^: upfield-shifted chemical shifts; ^b^: downfield shifted chemical shifts.

## Data Availability

Not applicable.

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
