# Peer review of "Densazalin, a New Cytotoxic Diazatricyclic Alkaloid from the Marine Sponge *Haliclona densaspicula"

_molecules, 2021, doi:10.3390/molecules26113164_

Round 1
Reviewer 1 Report
In this manuscript molecules-1220899, the authors reported a polycyclic alkaloid, densazalin, which was isolated from the marine sponge Haliclona densaspicula collected in Korea. This manuscript had the significance for the research of natural product. The manuscript mainly described the structure identification of the densazalin and its cell cytotoxic activity. Throughout the full manuscript, the major problem was that the X-ray of densazalin (extremely complex structure) was not acquired, which was related to the 2.3 mg of the compound. However, in my opinion, the manuscript could be worth published based on the data already obtained. Following are my main concerns:
1) In the “Introduction” part, the research status of the polycyclic alkaloids and their biological activity should be mentioned.
2) In the “Results and discussion” part, the IC50 should be obtained through fitted curve rather than through broken line.
3) The language should be improved.
Author Response
Thank you for your review with valuable comments. We tried to upgrade our manuscript based on your comments. As you said, X-ray determination for densazalin is important, but we did not try to make a crystal. This reason is that this compound might not be formed as a crystal due to the two flexible linear carbon chains.
1) In the “Introduction” part, the research status of the polycyclic alkaloids and their biological activity should be mentioned.
→ In accordance to Reviewer’s comments, “Introduction” has been improved. Polycyclic alkaloids from the genus Haliclona and their biological activities are additionally described.
2) In the “Results and discussion” part, the IC50 should be obtained through fitted curve rather than through broken line.
→ IC50 calculation was conducted from fitting curve. In the revised version, we replaced the curve in the figure 5 with smooth curve.
3) The language should be improved.
→ The English in the manuscript was corrected in the Editage English editing company.
Reviewer 2 Report
The manuscript presented for review contains interesting data. In my opinion, it can be published after a moderate revision.
Please consider the following:
- The introduction needs to be enhanced with supplementary relevant information.
- Materials and methods: please provide an image (in the Supplementary material) of the voucher specimen.
- Materials and methods: please provide details regarding analytical methods conditions.
- Results and discussion - please discuss the obtained results (regarding the cytotoxicity of densazalin) by comparison with literature data.
- References - most of the references used are rather old. Please introduce newer references (considering the points above)
Author Response
Thank you for your review with valuable comments. We tried to upgrade our manuscript based on your comments. And we responded to your points as below.
Please consider the following:
- The introduction needs to be enhanced with supplementary relevant information.
→ In accordance to Reviewer’s comments, “Introduction” has been improved. Polycyclic alkaloids from the genus Haliclona and their biological activities are additionally described.
- Materials and methods: please provide an image (in the Supplementary material) of the voucher specimen.
→ The image of H. densaspicula was included in the Supplementary material.
- Materials and methods: please provide details regarding analytical methods conditions.
→ We described the parameters for NMR measurements in Instruments part. And the isolation procedure was more detailed than the previous manuscript.
- Results and discussion - please discuss the obtained results (regarding the cytotoxicity of densazalin) by comparison with literature data.
→ Along with the reported data, we discussed our cytotoxicity
- References - most of the references used are rather old. Please introduce newer references (considering the points above)
→ Polycyclic alkaloids are unique compounds with very complex skeleton, hence the reports of a new compound in this group are very limited. In accordance to the reviewer’s comments the references regarding representative polycyclic alkaloids from the genus Haliclona are revised and added. Also, newer references regarding the biological activities of the genus Haliclona are added.
Reviewer 3 Report
The Authors reported on the isolation and complete molecular characterization of a new polycyclic alkaloid from the marine sponge Haliclona densaspicula with possible valuable application as cytotoxic compound to cure cancer.
IC50 values against both AGS and HepG2 cells were determined. These values should be compared with commercial anticancer drug standards. I suggest to add this comparison in the revision.
Moreover, these IC50 values are quite high in my opinion, but this suggest that this natural compound could constitute a new scaffold for the design of semisynthetic anticancer drugs. There are many example in the literature. This point should be added in the discussion.
Author Response
Reviewer 3
Thank you for your review with valuable comments. We tried to upgrade our manuscript based on your comments. And we responded to your points as below.
IC50 values against both AGS and HepG2 cells were determined. These values should be compared with commercial anticancer drug standards. I suggest to add this comparison in the revision.
→ In fact, we don’t think that the cytotoxicity of densazalin is not excellent, but moderate. Many marine sponges displayed the strong cytotoxicity: IC50 values are below10 uM. So we did not emphasize on cytotoxicity of densazalin. However, in accordance with your comment, we discussed our result compared with other strong marine alkaloids.
Moreover, these IC50 values are quite high in my opinion, but this suggest that this natural compound could constitute a new scaffold for the design of semisynthetic anticancer drugs. There are many example in the literature. This point should be added in the discussion.
→ This comment is similar to above point. We added the description of cytotoxicity in discussion.
Round 2
Reviewer 2 Report
The authors provided satisfactory responses to the questions raised by the reviewer in the first round of review.
The manuscript can be accepted in the present form.